# Interactive Label Cleaning with Example-based Explanations

**Stefano Teso**
University of Trento
Trento, Italy
stefano.teso@unitn.it

**Andrea Bontempelli**
University of Trento
Trento, Italy
andrea.bontempelli@unitn.it

**Fausto Giunchiglia**
University of Trento
Trento, Italy
fausto.giunchiglia@unitn.it

**Andrea Passerini**
University of Trento
Trento, Italy
andrea.passerini@unitn.it

## Abstract

We tackle sequential learning under label noise in applications where a human supervisor can be queried to relabel suspicious examples. Existing approaches are flawed, in that they only relabel incoming examples that look "suspicious" to the model. As a consequence, those mislabeled examples that elude (or don't undergo) this cleaning step end up tainting the training data and the model with no further chance of being cleaned. We propose CINCER, a novel approach that cleans both new and past data by identifying *pairs of mutually incompatible examples*. Whenever it detects a suspicious example, CINCER identifies a counter-example in the training set that—according to the model—is maximally incompatible with the suspicious example, and asks the annotator to relabel either or both examples, resolving this possible inconsistency. The counter-examples are chosen to be maximally incompatible, so to serve as *explanations* of the model's suspicion, and highly influential, so to convey as much information as possible if relabeled. CINCER achieves this by leveraging an efficient and robust approximation of influence functions based on the Fisher information matrix (FIM). Our extensive empirical evaluation shows that clarifying the reasons behind the model's suspicions by cleaning the counter-examples helps in acquiring substantially better data and models, especially when paired with our FIM approximation.

## 1   Introduction

Label noise is a major issue in machine learning as it can lead to compromised predictive performance and unreliable models [1, 2]. We focus on sequential learning settings in which a human supervisor, usually a domain expert, can be asked to double-check and relabel any potentially mislabeled example. Applications include crowd-sourced machine learning and citizen science, where trained researchers can be asked to clean the labels provided by crowd-workers [3, 4], and interactive personal assistants [5], where the user self-reports the initial annotations (e.g., about activities being performed) and unreliability is due to memory bias [6], unwillingness to report [7], or conditioning [8].

This problem is often tackled by monitoring for incoming examples that are likely to be mislabeled, *aka* suspicious examples, and ask the supervisor to provide clean (or at least better) annotations for them. Existing approaches, however, focus solely on cleaning the incoming examples [9, 4, 10, 5]. This means that noisy examples that did not undergo the cleaning step (e.g., those in the initial

35th Conference on Neural Information Processing Systems (NeurIPS 2021).

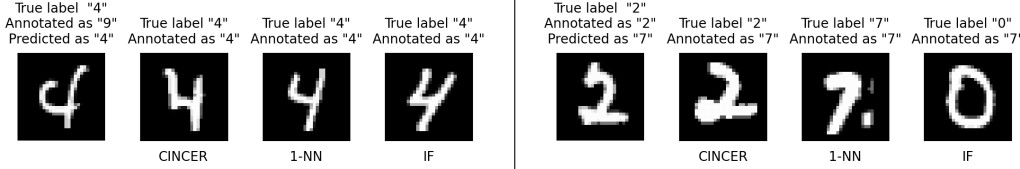

Figure 1: Suspicious example and counter-examples selected using (from left to right) CINCER, 1-NN and influence functions (IF), on noisy MNIST. **Left**: the suspicious example is mislabeled, the machine's suspicion is supported by a clean counter-example. **Right**: the suspicious example is not mislabeled, the machine is wrongly suspicious because the counter-example is mislabeled. CINCER's counter-example is contrastive and influential; 1-NN's is not influential and IF's is not pertinent, see desiderata D1–D3 below.

bootstrap data set) or that managed to elude it are left untouched. This degrades the quality of the model and prevents it from spotting future mislabeled examples that fall in regions affected by noise.

We introduce CINCER (Contrastive and InflueNt CounterExample stRategy), a new explainable interactive label cleaning algorithm that lets the annotator observe and fix the *reasons* behind the model's suspicions. For every suspicious example that it finds, CINCER identifies a counter-example, i.e., a training example that maximally supports the machine's suspicion. The idea is that the example/counter-example pair captures a potential inconsistency in the data—as viewed from the model's perspective—which is resolved by invoking the supervisor. More specifically, CINCER asks the user to relabel the example, the counter-example, or both, thus improving the quality of, and promoting consistency between, the data and the model. Two hypothetical rounds of interaction on a noisy version of MNIST are illustrated in Figure 1.

CINCER relies on a principled definition of counter-examples derived from few explicit, intuitive desiderata, using influence functions [11, 12]. The resulting counter-example selection problem is solved using a simple and efficient approximation based on the Fisher information matrix [13] that consistently outperforms more complex alternatives in our experiments.

**Contributions:** Summarizing, we: 1) Introduce CINCER, an explanatory interactive label cleaning strategy that leverages example-based explanations to identify inconsistencies in the data—as perceived by the model—and enable the annotator to fix them. 2) Show how to select counter-examples that at the same time explain why the model is suspicious and that are highly informative using (an efficient approximation of) influence functions. 3) Present an extensive empirical evaluation that showcases the ability of CINCER of building cleaner data sets and better models.

## 2  Background

We are concerned with sequential learning under label noise. In this setting, the machine receives a sequence of examples $\tilde{z}_t := (\mathbf{x}_t, \tilde{y}_t)$, for $t = 1, 2, \ldots$, where $\mathbf{x}_t \in \mathbb{R}^d$ is an instance and $\tilde{y}_t \in [c]$ is a corresponding label, with $[c] := \{1, \ldots, c\}$. The label $\tilde{y}_t$ is unreliable and might differ from the ground-truth label $y_t^*$. The key feature of our setting is that a *human supervisor* can be asked to double-check and relabel any example. The goal is to acquire a clean dataset and a high-quality predictor while asking few relabeling queries, so to keep the cost of interaction under control.

The state-of-the-art for this setting is skeptical learning (SKL) [10, 5]. SKL is designed primarily for smart personal assistants that must learn from unreliable users. SKL follows a standard sequential learning loop: in each iteration, the machine receives an example and updates the model accordingly. However, for each example that it receives, the machine compares (an estimate of) the quality of the annotation with that of its own prediction, and if the prediction looks more reliable than the annotation by some factor, SKL asks the user to double-check his/her example. The details depend on the implementation: in [10] label quality is estimated using the empirical accuracy for the classifier and the empirical probability of contradiction for the annotator, while in [5] the machine measures the margin between the user's and machine's labels. Our approach follows the latter strategy.

Another very related approach is learning from weak annotators (LWA) [9, 4], which focuses on querying domain experts rather than end-users. The most recent approach [4] jointly learns a

prediction pipeline composed of a classifier and a noisy channel, which allows it to estimate the noise rate directly. Moreover, the approach in [4] identifies suspicious examples that have a large impact on the learned model. A theoretical foundation for LWA is given in [9]. LWA is however designed for pool-based scenarios, where the training set is given rather than obtained sequentially. For this reason, in the remainder of the paper, we will chiefly build on and compare to SKL.

**Limitations of existing approaches.** A major downside of SKL is that it focuses on cleaning the incoming examples only. This means that if a mislabeled example manages to elude the cleaning step and gets added to the training set, it is bound to stay there forever. This situation is actually quite common during the first stage of skeptical learning, in which the model is highly uncertain and trusts the incoming examples—even if they are mislabeled. The same issue occurs if the initial training set used to bootstrap the classifier contains mislabeled examples. As shown by our experiments, the accumulation of noisy data in the training set may have a detrimental effect on the model's performance (cf. Figure 2). In addition, it can also affect the model's ability to identify suspicious examples: a noisy data point can fool the classifier into trusting incoming mislabeled examples that fall close to it, further aggravating the situation.

## 3   Explainable Interactive Label Cleaning with CINCER

We consider a very general class of probabilistic classifiers $f : \mathbb{R}^d \to [c]$ of the form $f(\mathbf{x}; \theta) :=$ $\operatorname{argmax}_{y \in [c]} \ P(y \mid \mathbf{x}; \theta)$, where the conditional distribution $P(Y \mid \mathbf{X}; \theta)$ has been fit on training data by minimizing the cross-entropy loss $\ell((\mathbf{x}, y), \theta) = -\sum_{i \in [c]} \mathbb{1}\{i = y\} \log P(i \mid \mathbf{x}, \theta)$. In our implementation, we also assume $P$ to be a neural network with a softmax activation at the top layer, trained using some variant of SGD and possibly early stopping.

### 3.1   The CINCER Algorithm

The pseudo-code of CINCER is listed in Algorithm 1. At the beginning of iteration $t$, the machine has acquired a training set $D_{t-1} = \{z_1, \ldots, z_{t-1}\}$ and trained a model with parameters $\theta_{t-1}$ on it. At this point, the machine receives a new, possibly mislabeled example $\tilde{z}_t$ (line 3) and has to decide whether to trust it.

Following skeptical learning [5], CINCER does so by computing the *margin* $\mu(\tilde{z}_t, \theta_{t-1})$, i.e., the difference in conditional probability between the model's prediction $\hat{y}_t := \operatorname{argmax}_y P(y \mid \mathbf{x}_t, \theta_{t-1})$ and the annotation $\tilde{y}_t$. More formally:

$$\mu(\tilde{z}_t, \theta_{t-1}) := P(\hat{y}_t \mid \mathbf{x}_t, \theta_{t-1}) - P(\tilde{y}_t \mid \mathbf{x}_t, \theta_{t-1}) \tag{1}$$

The margin estimates the incompatibility between the model and the example: the larger the margin, the more suspicious the example. The example $\tilde{z}_t$ is deemed compatible if the margin is below a given threshold $\tau$ and suspicious otherwise (line 4); possible choices for $\tau$ are discussed in Section 3.5.

If $\tilde{z}_t$ is compatible, it is added to the data set as-is (line 5). Otherwise, CINCER computes a counter-example $z_k \in D_{t-1}$ that maximally supports the machine's suspicion. The intuition is that the pair $(\tilde{z}_t, z_k)$ captures a potential *inconsistency* in the data. For instance, the counter-example might be a correctly labeled example that is close or similar to $\tilde{z}_t$ but has a different label, or a distant noisy outlier that fools the predictor into assigning low probability to $\tilde{y}_t$. How to choose an effective counter-example is a major focus of this paper and discussed in detail in Section 3.2 and following.

Next, CINCER asks the annotator to double-check the pair $(\tilde{z}_t, z_k)$ and relabel the suspicious example, the counter-example, or both, thus resolving the potential inconsistency. The data set and model are then updated accordingly (line 9) and the loop repeats.

### 3.2   Counter-example Selection

Counter-examples are meant to illustrate why a particular example $\tilde{z}_t$ is deemed suspicious by the machine in a way that makes it easy to elicit useful corrective feedback from the supervisor. We posit that a good counter-example $z_k$ should be:

   D1. *Contrastive*: $z_k$ should explain why $\tilde{z}_t$ is considered suspicious by the model, thus highlighting a potential inconsistency in the data.

**Algorithm 1** Pseudo-code of CINCER. **Inputs**: initial (noisy) training set $D_0$; threshold $\tau$.

---

1: fit $\theta_0$ on $D_0$
2: **for** $t = 1, 2, \ldots$ **do**
3:     receive new example $\tilde{z}_t = (\mathbf{x}_t, \tilde{y}_t)$
4:     **if** $\mu(\tilde{z}_t, \theta_{t-1}) < \tau$ **then**
5:         $D_t \leftarrow D_{t-1} \cup \{\tilde{z}_t\}$                               $\triangleright \tilde{z}_t$ is compatible
6:     **else**
7:         find counterexample $z_k$ using Eq. 12                    $\triangleright \tilde{z}_t$ is suspicious
8:         present $\tilde{z}_t, z_k$ to the user, receive possibly cleaned labels $y'_t, y'_k$
9:         $D_t \leftarrow (D_{t-1} \setminus \{z_k\}) \cup \{(\mathbf{x}_t, y'_t), (\mathbf{x}_k, y'_k)\}$
10:    fit $\theta_t$ on $D_t$

---

D2. *Influential*: if $z_k$ is mislabeled, correcting it should improve the model as much as possible, so to maximize the information gained by interacting with the annotator.

In the following, we show how, for models learned by minimizing the cross-entropy loss, one can identify counter-examples that satisfy *both* desiderata.

**What is a contrastive counter-example?** We start by tackling the first desideratum. Let $\theta_{t-1}$ be the parameters of the current model. Intuitively, $z_k \in D_{t-1}$ is a contrastive counter-example for a suspicious example $\tilde{z}_t$ if *removing* it from the data set and retraining leads to a model with parameters $\theta_{t-1}^{-k}$ that assigns *higher* probability to the suspicious label $\tilde{y}_t$. The most contrastive counter-example is then the one that maximally affects the change in probability:

$$\text{argmax}_{k \in [t-1]} \ \left\{ P(\tilde{y}_t \mid \mathbf{x}_t; \theta_{t-1}^{-k}) - P(\tilde{y}_t \mid \mathbf{x}_t; \theta_{t-1}) \right\} \tag{2}$$

While intuitively appealing, optimizing Eq. 2 directly is computationally challenging as it involves retraining the model $|D_{t-1}|$ times. This is impractical for realistically sized models and data sets, especially in our interactive scenario where a counter-example must be computed in each iteration.

**Influence functions.** We address this issue by leveraging influence functions (IFs), a computational device that can be used to estimate the impact of specific training examples without retraining [11, 12]. Let $\theta_t$ be the empirical risk minimizer on $D_t$ and $\theta_t(z, \epsilon)$ be the minimizer obtained after adding an example $z$ with weight $\epsilon$ to $D_t$, namely:

$$\theta_t := \text{argmin}_\theta \frac{1}{t} \sum_{k=1}^t \ell(z_k, \theta) \qquad \theta_t(z, \epsilon) := \text{argmin}_\theta \frac{1}{t} \left( \sum_{k=1}^t \ell(z_k, \theta) \right) + \epsilon \ell(z, \theta) \tag{3}$$

Taking a first-order Taylor expansion, the difference between $\theta_t = \theta_t(z, 0)$ and $\theta_t(z, \epsilon)$ can be written as $\theta_t(z, \epsilon) - \theta_t(z, 0) \approx \epsilon \cdot \left( \frac{d}{d\epsilon} \theta_t(z, \epsilon) \big|_{\epsilon=0} \right)$. The derivative appearing on the right hand side is the so-called influence function, denoted $\mathcal{I}_{\theta_t}(z)$. It follows that the effect on $\theta_t$ of adding (resp. removing) an example $z$ to $D_t$ can be approximated by multiplying the IF by $\epsilon = 1/t$ (resp. $\epsilon = -1/t$). Crucially, if the loss is strongly convex and twice differentiable, the IF can be written as:

$$\mathcal{I}_{\theta_t}(z) = -H(\theta_t)^{-1} \nabla_\theta \ell(z, \theta_t) \tag{4}$$

where the curvature matrix $H(\theta_t) := \frac{1}{t} \sum_{k=1}^t \nabla_\theta^2 \ell(z_k, \theta_t)$ is positive definite and invertible. IFs were shown to capture meaningful information even for neural networks and other non-convex models [12].

**Identifying contrastive counter-examples with influence functions.** To see the link between contrastive counter-examples and influence functions, notice that the second term of Eq. 2 is independent of $z_k$, while the first term can be conveniently approximated with IFs by applying the chain rule:

$$-\frac{1}{t-1} \left( \frac{d}{d\epsilon} P(\tilde{y}_t \mid \mathbf{x}_t; \theta_{t-1}(z_k, \epsilon)) \Big|_{\epsilon=0} \right) = -\frac{1}{t-1} \left( \nabla_\theta P(\tilde{y}_t \mid \mathbf{x}_t; \theta_{t-1})^\top \frac{d}{d\epsilon} \theta_{t-1}(z_k, \epsilon) \Big|_{\epsilon=0} \right) \tag{5}$$

$$= -\frac{1}{t-1} \nabla_\theta P(\tilde{y}_t \mid \mathbf{x}_t; \theta_{t-1})^\top \mathcal{I}_{\theta_{t-1}}(z_k) \tag{6}$$

The constant can be dropped during the optimization. This shows that Eq. 2 is equivalent to:

$$\text{argmax}_{k \in [t-1]} \ \nabla_\theta P(\tilde{y}_t \mid \mathbf{x}_t; \theta_{t-1})^\top H(\theta_{t-1})^{-1} \nabla_\theta \ell(z_k, \theta_{t-1}) \tag{7}$$

Eq. 7 can be solved efficiently by combining two strategies [12]: i) Caching the inverse Hessian-vector product (HVP) $\nabla_\theta P(\tilde{y}_t \mid \mathbf{x}_t; \theta_{t-1})^\top H(\theta_{t-1})^{-1}$, so that evaluating the objective on each $z_k$ becomes a simple dot product, and ii) Solving the inverse HVP with an efficient stochastic estimator like LISSA [14]. This gives us an algorithm for computing contrastive counter-examples.

**Contrastive counter-examples are highly influential.** Can this algorithm be used for identifying influential counter-examples? It turns out that, as long as the model is obtained by optimizing the cross-entropy loss, the answer is affirmative. Indeed, note that if $\ell(z, \theta) = -\log P(y \mid \mathbf{x}; \theta)$, then:

$$\nabla_\theta P(\tilde{y}_t \mid \mathbf{x}_t; \theta_{t-1}) = P(\tilde{y}_t \mid \mathbf{x}_t; \theta_{t-1}) \frac{\nabla_\theta P(\tilde{y}_t \mid \mathbf{x}_t; \theta_{t-1})}{P(\tilde{y}_t \mid \mathbf{x}_t; \theta_{t-1})} = \tag{8}$$

$$= P(\tilde{y}_t \mid \mathbf{x}_t; \theta_{t-1}) \nabla_\theta \log P(\tilde{y}_t \mid \mathbf{x}_t; \theta_{t-1}) = -P(\tilde{y}_t \mid \mathbf{x}_t; \theta_{t-1}) \nabla_\theta \ell(\tilde{z}_t, \theta_{t-1}) \tag{9}$$

Hence, Eq. 6 can be rewritten as:

$$- P(\tilde{y}_t \mid \mathbf{x}_t; \theta_{t-1}) \nabla_\theta \ell(\tilde{z}_t, \theta_{t-1})^\top H(\theta_{t-1})^{-1} \nabla_\theta \ell(z_k, \theta_{t-1}) \tag{10}$$

$$\propto -\nabla_\theta \ell(\tilde{z}_t, \theta_{t-1})^\top H(\theta_{t-1})^{-1} \nabla_\theta \ell(z_k, \theta_{t-1}) \tag{11}$$

It follows that, under the above assumptions and as long as the model satisfies $P(\tilde{y}_t \mid \mathbf{x}_t; \theta_{t-1}) > 0$, Eq. 2 is equivalent to:

$$\operatorname{argmax}_{k \in [t-1]} \ -\nabla_\theta \ell(\tilde{z}_t, \theta_{t-1})^\top H(\theta_{t-1})^{-1} \nabla_\theta \ell(z_k, \theta_{t-1}) \tag{12}$$

This equation recovers *exactly* the definition of influential examples given in [12, Eq. 2] and shows that, for the large family of classifiers trained by cross-entropy, highly influential counter-examples are highly contrastive and vice versa, so that no change to the selection algorithm is necessary.

## 3.3   Counter-example Selection with the Fisher information matrix

Unfortunately, we found the computation of IFs to be unreliable in practice, cf. [15]. This leads to unstable ranking of candidates and reflects on the quality of the counter-examples, as in Figure 1. The issue is that, for non-convex classifiers trained using gradient-based methods (and possibly early stopping), $\theta_{t-1}$ is seldom close to a local minimum, rendering the Hessian non-positive definite. In our setting, the situation is further complicated by the presence of noise, which dramatically skews the curvature of the empirical risk. Remedies like fine-tuning the model with L-BFGS [12, 16], preconditioning and weight decay [15] proved unsatisfactory in our experiments.

We take a different approach. The idea is to replace the Hessian by the Fisher information matrix (FIM). The FIM $F(\theta)$ of a discriminative model $P(Y \mid \mathbf{X}, \theta)$ and training set $D_{t-1}$ is [17, 18]:

$$F(\theta) := \frac{1}{t-1} \sum_{k=1}^{t-1} \mathbb{E}_{y \sim P(Y \mid \mathbf{x}_k, \theta)} \left[ \nabla_\theta \log P(y \mid \mathbf{x}_k, \theta) \nabla_\theta \log P(y \mid \mathbf{x}_k, \theta)^\top \right] \tag{13}$$

It can be shown that, if the model approximates the data distribution, the FIM approximates the Hessian, cf. [19, 20]. Even when this assumption does not hold, as is likely in our noisy setting, the FIM still captures much of the curvature information encoded into the Hessian [17]. Under this approximation, Eq. 12 can be rewritten as:

$$\operatorname{argmax}_{k \in [t-1]} \ -\nabla_\theta \ell(\tilde{z}_t, \theta_{t-1})^\top F(\theta_{t-1})^{-1} \nabla_\theta \ell(z_k, \theta_{t-1}) \tag{14}$$

The advantage of this formulation is twofold. First of all, this optimization problem also admits caching the inverse FIM-vector product (FVP), which makes it viable for interactive usage. Second, and most importantly, the FIM is positive semi-definite by construction, making the computation of Eq.14 much more stable.

The remaining step is how to compute the inverse FVP. Naïve storage and inversion of the FIM, which is $|\theta| \times |\theta|$ in size, is impractical for typical models, so the FIM is usually replaced with a simpler matrix. Three common options are the identity matrix, the diagonal of the FIM, and a block-diagonal approximation where interactions between parameters of different layers are set to zero [17]. Our best results were obtained by restricting the FIM to the top layer of the network. We refer to this approximation as "Top Fisher". While more advanced approximations like K-FAC [17] exist, the Top Fisher proved surprisingly effective in our experiments.

### 3.4 Selecting Pertinent Counter-examples

So far, we have discussed how to select contrastive and influential counter-examples. Now we discuss how to make the counter-examples easier to interpret for the annotator. To this end, we introduce the additional desideratum that counter-examples should be:

D3 *Pertinent*: it should be clear *to the user* why $z_k$ is a counter-example for $\tilde{z}_t$.

We integrate D3 into CINCER by restricting the choice of possible counter-examples. A simple strategy, which we do employ in all of our examples and experiments, is to restrict the search to counter-examples whose label in the training set is the same as the prediction for the suspicious example, i.e., $y_k = \hat{y}_t$. This way, the annotator can interpret the counter-example as being in support of the machine's suspicion. In other words, if the counter-example is labeled correctly, then the machine's suspicion is likely right and the incoming example needs cleaning. Otherwise, if the machine is wrong and the suspicious example is not mislabeled, it is likely the counter-example – which backs the machine's suspicions – that needs cleaning.

Finally, one drawback of IF-selected counter-examples is that they may be perceptually different from the suspicious example. For instance, outliers are often highly influential as they fool the machine into mispredicting many examples, yet they have little in common with those examples [20]. This can make it difficult for the user to understand their relationship with the suspicious examples they are meant to explain. This is not necessarily an issue: first, a motivated supervisor is likely to correct mislabeled counter-examples regardless of whether they resemble the suspicious example; second, highly influential outliers are often identified (and corrected if needed) in the first iterations of CINCER (indeed, we did not observe a significant amount of repetitions among suggested counter-examples in our experiments). Still, CINCER can be readily adapted to acquire more perceptually similar counter-examples. One option is to replace IFs with *relative* IFs [20], which trade-off influence with locality. Alas, the resulting optimization problem does not support efficient caching of the inverse HVP. A better alternative is to restrict the search to counter-examples $z_k$ that are similar enough to $\tilde{z}_t$ in terms of some given perceptual distance $\|\cdot\|_\mathcal{P}$ [21] by filtering the candidates using fast nearest neighbor techniques in perceptual space. This is analogous to FastIF [22], except that the motivation is to encourage perceptual similarity rather than purely efficiency, although the latter is a nice bonus.

### 3.5 Advantages and Limitations

The main benefit of CINCER is that, by asking a human annotator to correct potential inconsistencies in the data, it acquires substantially better supervision and, in turn, better predictors. In doing so, CINCER also encourages consistency between the data and the model. Another benefit is that, by explaining the reasons behind the model's skepticism, CINCER allows the supervisor to spot bugs and justifiably build – or, perhaps more importantly, reject [23, 24] – trust into the prediction pipeline.

CINCER only requires to set a single parameter, the margin threshold $\tau$, which determines how frequently the supervisor is invoked. The optimal value is highly application-specific, but generally speaking, it depends on the ratio between the cost of a relabeling query and the cost of noise. If the annotator is willing to interact (for instance, because it is paid to do so) then the threshold can be quite generous.

## 4 Experiments

We empirically address the following research questions: **Q1**: Do counter-examples contribute to cleaning the data? **Q2**: Which influence-based selection strategy identifies the most mislabeled counter-examples? **Q3**: What contributes to the effectiveness of the best counter-example selection strategy?

We implemented CINCER using Python and Tensorflow [25] on top of three classifiers and compared different counter-example selection strategies on five data sets. The IF code is adapted from [26]. All experiments were run on a 12-core machine with 16 GiB of RAM and no GPU. The code for all experiments is available at: `https://github.com/abonte/cincer`.

**Data sets.** We used a diverse set of classification data sets: **Adult** [27]: data set of 48,800 persons, each described by 15 attributes; the goal is to discriminate customers with an income above/below

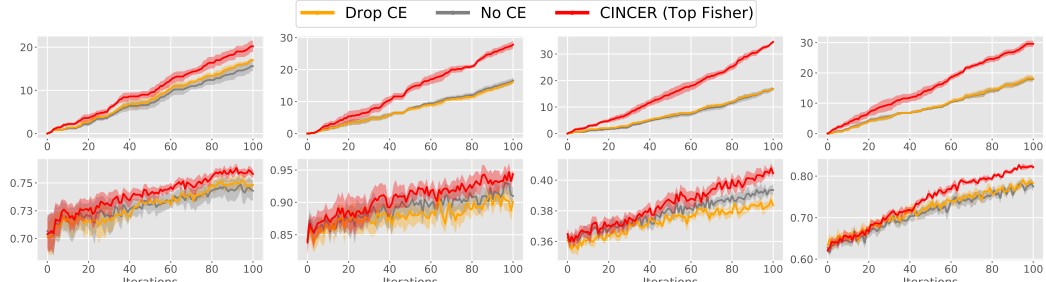

Figure 2: CINCER using Top Fisher vs. drop CE and no CE. **Left to right**: results for FC on adult, breast and 20NG, CNN on MNIST. **Top row**: # of cleaned examples. **Bottom row**: $F_1$ score.

$50K. **Breast** [27]: data set of 569 patients described by 30 real-valued features. The goal is to discriminate between benign and malignant breast cancer cases. **20NG** [27]: data set of newsgroup posts categorized in twenty topics. The documents were embedded using a pre-trained Sentence-BERT model [28] and compressed to 100 features using PCA. **MNIST** [29]: handwritten digit recognition data set from black-and-white, $28 \times 28$ images with pixel values normalized in the $[0, 1]$ range. The data set consists of 60K training and 10K test examples. **Fashion** [30]: fashion article classification dataset with the same structure as MNIST. For adult and breast, a random $80 : 20$ training-test split is used while for MNIST, fashion and 20 NG the split provided with the data set is used. The labels of 20% of training examples were *corrupted* at random. The experiments were repeated five times, each time changing the seed used for corrupting the data. Performance was measured in terms of $F_1$ score on the (uncorrupted) test set. Error bars in the plots indicate the standard error. All competitors received exactly the same examples in exactly the same order.

**Models.** We applied CINCER to three models: **LR**, a logistic regression classifier; **FC**, a feed-forward neural network with two fully connected hidden layers with ReLU activations; and **CNN**, a feed-forward neural network with two convolutional layers and two fully connected layers. For all models, the hidden layers have ReLU activations and 20% dropout while the top layer has a softmax activation. LR was applied to MNIST, FC to both the tabular data sets (namely: adult, breast, german, and 20NG) and image data sets (MNIST and fashion), and CNN to the image data sets only. Upon receiving a new example, the classifier is retrained from scratch for 100 epochs using Adam [31] with default parameters, with early stopping when the accuracy on the training set reaches 90% for FC and CNN, and 70% for LR. This helps substantially to stabilize the quality of the model and speeds up the evaluation. Before each run, the models are trained on a bootstrap training set (containing 20% mislabeled examples) of 500 examples for 20NG and 100 for all the other data sets. The margin threshold is set to $\tau = 0.2$. Due to space constraints, we report the results on one image data set and three tabular data, and we focus on FC and CNN. The other results are consistent with what is reported below; these plots are reported in the Supplementary Material.

### 4.1 Q1: Counter-examples improve the quality of the data

To evaluate the impact of cleaning the counter-examples, we compare CINCER combined with the Top fisher approximation of the FIM, which works best in practice, against two alternatives, namely: **No CE**: an implementation of skeptical learning [5] that asks the user to relabel any incoming suspicious examples identified by the margin and presents no counter-examples. **Drop CE**: a variation of CINCER that identifies counter-examples using Top Fisher but drops them from the data set if the user considers the incoming example correctly labeled. The results are reported in Figure 2. The plots show that CINCER cleans by far the most examples on all data sets, between 33% and 80% more than the alternatives (top row in Figure 2). This translates into better predictive performance as measured by $F_1$ score (bottom row). Notice also that CINCER consistently outperforms the drop CE strategy in terms of $F_1$ score, suggesting that relabeling the counter-examples provides important information for improving the model. These results validate our choice of identifying and relabeling counter-examples for interactive label cleaning compared to focusing on suspicious incoming examples only, and allow us to answer **Q1** in the affirmative.

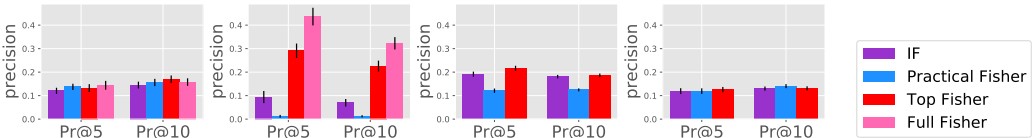

Figure 3: Counter-example Pr@5 and Pr@10. Standard error information is reported. **Left to right**: results for FC on adult, breast and 20NG, and CNN on MNIST.

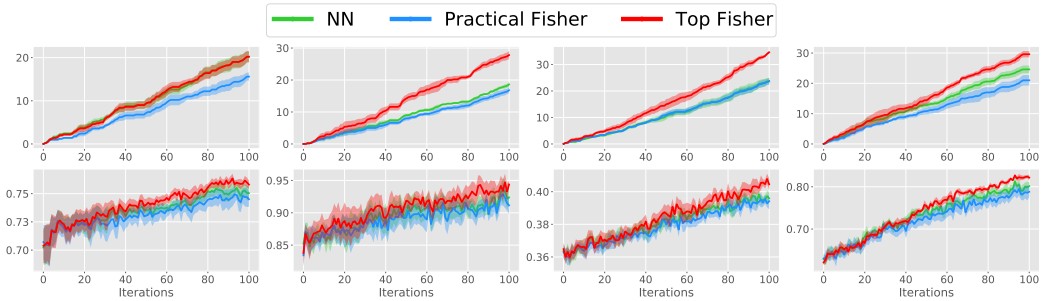

Figure 4: Top Fisher vs. practical Fisher vs. NN. **Left to right**: results for FC on adult, breast and 20NG, CNN on MNIST. **Top row**: # of cleaned examples. **Bottom row**: $F_1$ score.

### 4.2 Q2: Fisher Information-based strategies identify the most mislabeled counter-examples

Next, we compare the ability of alternative approximations of IFs to discover mislabeled counter-examples. To this end, we trained a model on a noisy bootstrap data set, selected 100 examples from the remainder of the training set, and measured how many truly mislabeled counter-examples are selected by alternative strategies. In particular, we computed influence using the IF LISSA estimator of [12], the actual FIM (denoted "full Fisher" and reported for the simpler models only for computational reasons) and its approximations using the identity matrix (*aka* "practical Fisher" [32]), and Top Fisher. We computed the precision at $k$ for $k \in \{5, 10\}$, i.e, the fraction of mislabeled counter-examples within five or ten highest-scoring counter-examples retrieved by the various alternatives, averaged over 100 iterations for five runs. The results in Figure 3 show that, in general, FIM-based strategies outperform the LISSA estimator, with Full Fisher performing best and Top Fisher a close second. Since the full FIM is highly impractical to store and invert, this confirms our choice of Top Fisher as the best practical strategy.

### 4.3 Q3: Both influence and curvature contribute to the effectiveness of Top Fisher

Finally, we evaluate the impact of selecting counter-examples using Top Fisher on the model's performance, in terms of use of influence, by comparing it to an intuitive nearest neighbor alternative (**NN**), and modelling of the curvature, by comparing it to the Practical Fisher. **NN** simply selects the counter-example that is closest to the suspicious example. The results can be viewed in Figure 4. Top Fisher is clearly the best strategy, both in terms of number of cleaned examples and $F_1$ score. **NN** is always worse than Top Fisher in terms of $F_1$, even in the case of adult (first column) when it cleans the same number of examples, confirming the importance of influence in selecting impactful counter-examples. Practical Fisher clearly underperforms compared with Top Fisher, and it shows the importance of having the curvature matrix. For each data set, all methods make a similar number of queries: 58 for 20NG, 21 for breast, 31 for adult and 37 for MNIST. In general, CINCER detects around 75% of the mislabeled examples (compared to 50% of the other methods) and only about 5% of its queries do not involve a corrupted example or counter-example. The complete values are reported in the Supplementary Material. As a final remark, we note that CINCER cleans more suspicious examples than counter-examples (in a roughly $2 : 1$ ratio), as shown by the number of cleaned suspicious examples vs. counter-examples reported in the Supplementary Material. Comparing this to the curve for Drop CE shows that proper data cleaning improves the ability of the model of being suspicious for the right reasons, as expected.

# 5 Related Work

**Learning under noise.** Typical strategies to learning from noisy labels include discarding or down-weighting suspicious examples and employing models robust to noise [33, 1, 34, 2], often requiring a non-trivial noise ratio estimation step [35]. These approaches make no attempt to recover the ground-truth label and are not ideal in interactive learning settings characterized by high noise rate/cost and small data sets. Most works on interactive learning under noise are designed for crowd-sourcing applications in which items are labelled by different annotators of varying quality and the goal is to aggregate weak annotations into a high-quality consensus label [3]. Our work is strongly related to approaches to interactive learning under label noise like skeptical learning [10, 5] and learning from weak annotators [9, 4]. These approaches completely ignore the issue of noise in the training set, which can be quite detrimental, as shown by our experiments. Moreover, they are completely black-box and do not attempt to explain to the supervisor why examples are considered suspicious by the machine, making it hard for him/her to establish or reject trust in the data and the model.

**Influence functions and Fisher information.** It is well known that mislabeled examples tend to exert a larger influence on the model [36, 12, 37, 20] and indeed IFs may be a valid alternative to the margin for identifying suspicious examples. Building on the seminal work of Koh and Liang [12], we instead leverage IFs to define and compute *contrastive* counter-examples that explain why the machine is suspicious. The difference is that noisy training examples influence the model as a whole, whereas contrastive counter-examples influence a specific suspicious example. To the best of our knowledge, this application of IFs is entirely novel. Notice also that empirical evidence that IFs recover noisy examples is restricted to offline learning [12, 37]. Our experiments extend this to a less forgiving interactive setting in which only one counter-example is selected per iteration and the model is trained on the whole training set. The idea of exploiting the FIM to approximate the Hessian has ample support in the natural gradient descent literature [17, 18]. The FIM has been used for computing example-based explanations by Khanna *et al.* [37]. However, their approach is quite different from ours. CINCER is equivalent to maximizing the Fisher kernel [32] between the suspicious example and the counter-example (Eq. 14) for the purpose of explaining the model's margin, and this formulation is explicitly derived from two simple desiderata. In contrast, Khanna *et al.* maximize a *function* of the Fisher kernel (namely, the *squared* Fisher kernel between $z_k$ and $\tilde{z}_t$ divided by the norm of $z_k$ in the RKHS). This optimization problem is not equivalent to Eq. 14 and does not admit efficient computation by caching the inverse FIM-vector product.

**Other works.** CINCER draws inspiration from explanatory active learning, which integrates local [24, 38, 39, 40] or global [41] explanations into interactive learning and allows the annotator to supply corrective feedback on the model's explanations. These approaches differ from CINCER in that they neither consider the issue of noise nor perform label cleaning, and indeed they explain the model's *predictions* rather than the model's *suspicion*. Another notable difference is that they rely on attribution-based explanations, whereas the backbone of CINCER is example-based explanations, which enable users to reason about labels in terms of concrete (training) examples [42, 43]. Following these works, saliency maps – which provide complementary information about relevant attributes – could potentially be integrated into CINCER to provide more fine-grained explanations and control.

# 6 Conclusion

We introduced CINCER, an approach for handling label noise in sequential learning that asks a human supervisor to relabel any incoming suspicious examples. Compared to previous approaches, CINCER identifies the reasons behind the model's skepticism and asks the supervisor to double-check them too. This is done by computing a training example that maximally supports the machine's suspicions. This enables the user to correct both incoming and old examples, cleaning inconsistencies in the data that confuse the model. Our experiments shows that, by removing inconsistencies in the data, CINCER enables acquiring better data and models than less informed alternatives.

Our work can be improved in several ways. CINCER can be straightforwardly extended to online active and skeptical learning, in which the label of incoming instances is acquired on the fly [44, 10]. CINCER can also be adapted to correcting multiple counter-examples as well as the reasons behind mislabeled counter-examples using "multi-round" label cleaning and group-wise measures of influence [45, 46, 47]. This more refined strategy is especially promising for dealing with systematic noise, in which counter-examples are likely affected by entire groups of mislabeled examples.

**Potential negative impact.** Like most interactive approaches, there is a risk that CINCER annoys the user by asking an excessive number of questions. This is currently mitigated by querying the user only when the model is confident enough in its own predictions (through the margin-based strategy) and by selecting influential counter-examples that have a high chance to improve the model upon relabeling, thus reducing the future chance of pointless queries. Moreover, the margin threshold $\tau$ allows to modulate the amount of interaction based on the user's commitment. Another potential issue is that CINCER could give malicious annotators fine-grained control over the training data, possibly leading to poisoning attacks. This is however not an issue for our target applications, like interactive assistants, in which the user benefits from interacting with a high-quality predictor and is therefore motivated to provide non-adversarial labels.

## Acknowledgments and Disclosure of Funding

This research has received funding from the European Union's Horizon 2020 FET Proactive project "WeNet - The Internet of us", grant agreement No. 823783, and from the "DELPhi - DiscovEring Life Patterns" project funded by the MIUR Progetti di Ricerca di Rilevante Interesse Nazionale (PRIN) 2017 – DD n. 1062 del 31.05.2019. The research of ST and AP was partially supported by TAILOR, a project funded by EU Horizon 2020 research and innovation programme under GA No 952215.

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
