# Interactive Label Cleaning with Example-based Explanations

**Stefano Teso**
University of Trento
Trento, Italy
stefano.teso@unitn.it

**Andrea Bontempelli**
University of Trento
Trento, Italy
andrea.bontempelli@unitn.it

**Fausto Giunchiglia**
University of Trento
Trento, Italy
fausto.giunchiglia@unitn.it

**Andrea Passerini**
University of Trento
Trento, Italy
andrea.passerini@unitn.it

## A Additional Details

For all models and data sets, the margin threshold is set to $\tau = 0.2$, the batch size is 1024 and the number of epochs is 100. As for influence functions, we made use of an implementation based on LISSA-based strategy suggested by Koh and Liang [1]. The Hessian damping (pre-conditioning) constant was set to 0.01, the number of stochastic LISSA iterations to 10 and the number of samples to 1 (the default value). We experimented with a large number of alternative hyperaparameter settings, including larger number of LISSA iterations (up to 1000) and number of samples (up to 30), without any substantial improvements in performance for the IF approximation.

## B Full Plots for Q1

Figure 1 reports the total number of cleaned examples (solid lines) and cleaned counter-examples (dashed lines), $F_1$ score and number of queries to the user. The results are the same as in the main text: CINCER combined with the Top Fisher approximation of the FIM is by far the best performing method. In all cases, CINCER cleans more examples and outperforms in terms of $F_1$ the alternative approaches for noise handling, namely drop CE and no CE. The number of cleaned counter-examples across data sets and models is more than 30% of the total number of cleaned examples. By comparing the curve of the cleaned counter-examples of CINCER with the the curve of Drop CE, we note that proper data cleaning improves the model's ability to be suspicious for the right reasons. The number of queries of all methods is similar across the data sets and models with few queries of difference. The useless queries, which do not contain at least one corrupted example or counter-example, are around 5% of the queries. For the same number of queries, CINCER cleans more labels confirming the advantages of identifying and relabeling counter-examples to increase the predictive performance.

## C Full Plots for Q2

To compare the number of mislabeled counter-examples discovered by the different approximation of IFs, we compute the precision at $k$ for $k \in \{5, 10\}$. The results are shown in Figure 2. In general, FIM-based approaches outperform the LISSA estimator. Top Fisher is the best strategy after the full FIM, which is difficult to store and invert.

35th Conference on Neural Information Processing Systems (NeurIPS 2021).

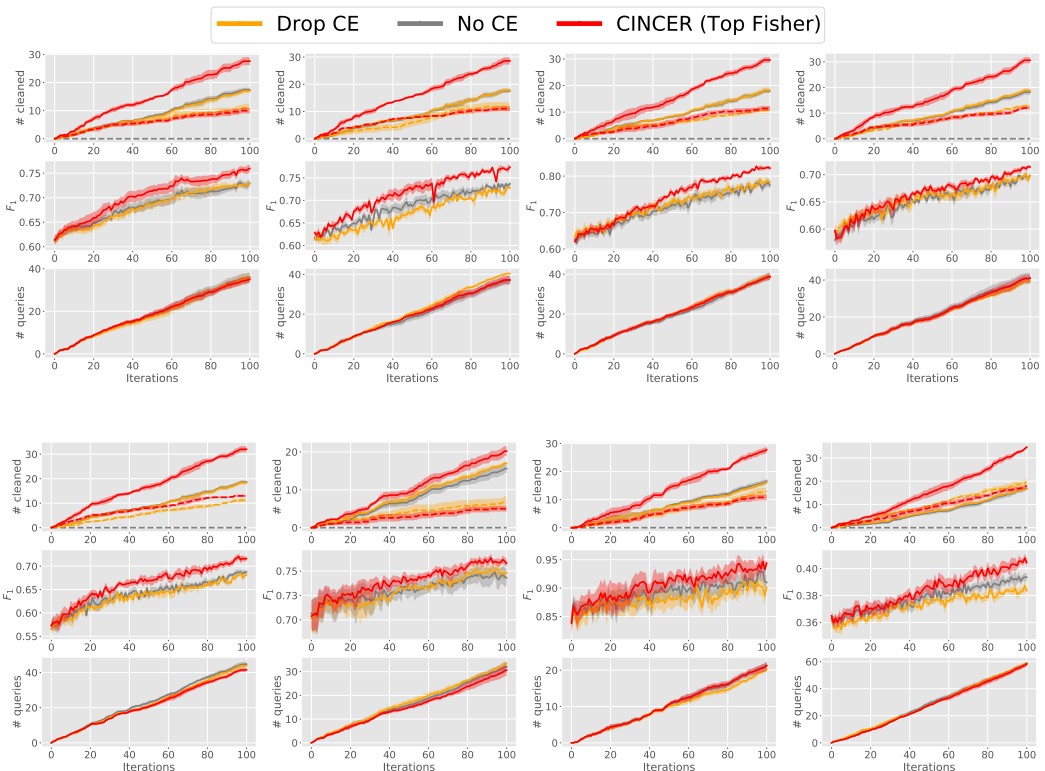

Figure 1: CINCER using Top Fisher vs. drop CE and no CE. **Top row from left to right**: LR, FC, CNN on MNIST and FC on fashion MNIST. **Bottom row from left to right**: CNN on fashion MNIST, FC on adult, breast and 20NG. **For each row**: total number of cleaned examples (solid lines) and counter-examples (dashed lines), $F_1$ score and number of queries.

## D  Full Plots for Q3

Figure 3 shows the results of the evaluation of Top Fisher, Practical Fisher and nearest neighbor (NN). Top Fisher outperforms the alternatives in terms of $F_1$ score and number of cleaned examples. NN is always worse than Top Fisher, even on adult (second column, second row) where it cleans the same number of examples but achieves lower predictive performance. These results show the importance of using the influence for choosing the counter-examples. CINCER identifies more corrupted counter-examples (in one case, the same number) than the other strategies, showing the advantage of using Top Fisher. As reported in the main text, Practical Fisher lags behind Top Fisher in all cases. The number of queries is similar for all strategies.

## References

[1] Pang Wei Koh and Percy Liang. Understanding black-box predictions via influence functions. In *Proceedings of the 34th International Conference on Machine Learning*, pages 1885–1894. JMLR. org, 2017.

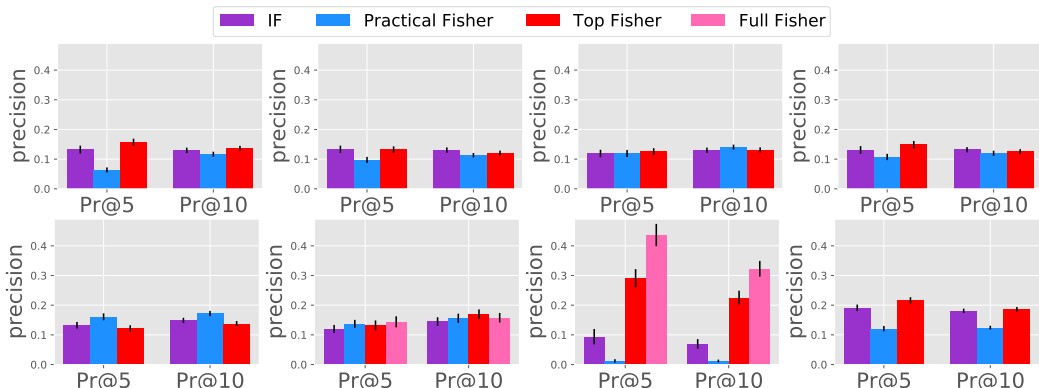

Figure 2: Counter-example Pr@5 and Pr@10. Standard error is reported. **Top row from left to right**: LR, FC, CNN on MNIST and FC on fasion MNIST. **Bottom row from left to right**: CNN on fashion MNIST, FC on adult, breast and 20NG.

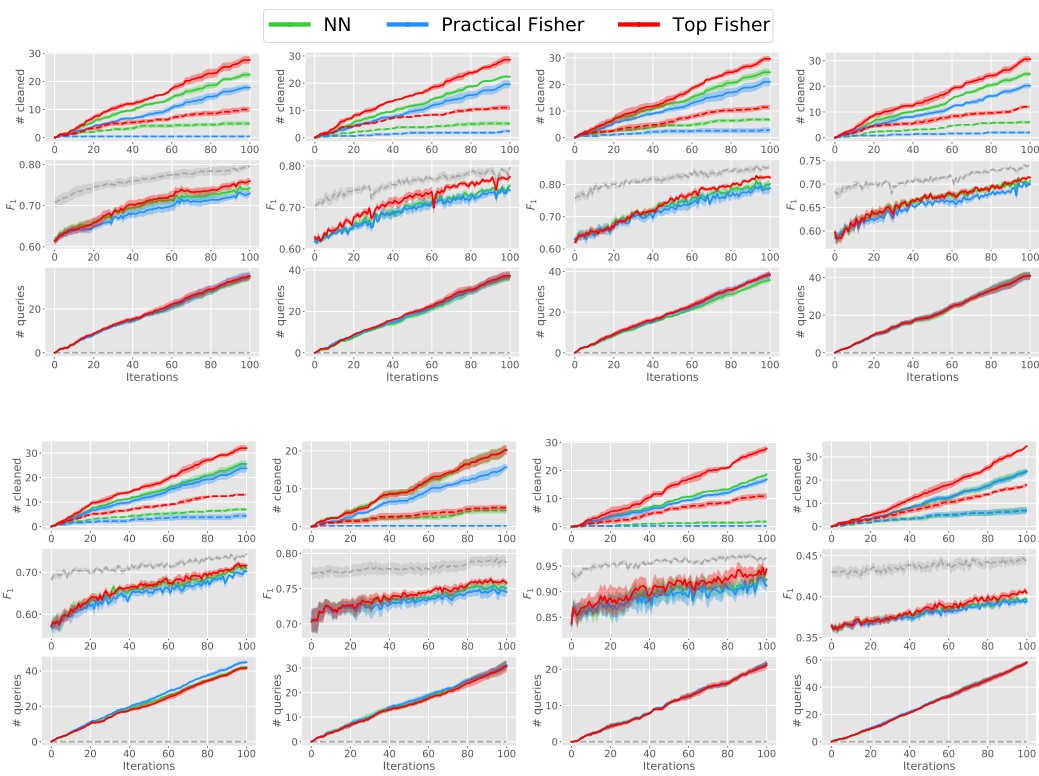

Figure 3: Top Fisher vs. practical Fisher vs. NN. **Top row from left to right**: LR, FC, CNN on MNIST and FC on fashion MNIST. **Bottom row from left to right**: CNN on fashion, FC on adult, breast and 20NG. **For each row**: total number of cleaned examples (solid lines) and counter-examples (dashed lines), $F_1$ score and number of queries. The dashed grey lines in the $F_1$ score are the upper bound, i.e., labels are not corrupted.