# OpenReview forum: "Interactive Label Cleaning with Example-based Explanations"
_NeurIPS.cc/2021/Conference — NeurIPS 2021 Spotlight_

### Official Review · Reviewer_CVDx · 2021-07-15

**Rating:** 7
**Confidence:** 3

**Summary:**

The article introduces a novel approach to interactively cleaning the erroneous labels for an ML task. It works by identifying highly-influential, mutually-incompatible examples. The authors show that, over time, correcting the newly acquired example (or the incompatible old one, or both), does improve the accuracy of the learned model.

**Main Review:**

The authors propose a novel approach to an important problem: improving the learned models by detecting and leaning the labels of mis-labeled examples. The paper is well written, easy to follow, and makes a compelling case for the proposed approach. The contribution appears to be original. The authors make a strong case for the use of influence functions in conjunction with a few explicit criteria (contrastive, influential, and pertinent).

In this reviewer's opinion, the paper could be improved by a few additions:
- adding an error analysis section: what proportion of the mislabeled examples can you detect? what proportion of the detected  contrastive pairs do NOT contain at least one example for which you did NOT corrupt the label?
- adding to Figures 2 & 4 the upper-bound performance of the model (ie, if it were trained w/o any corrupted labels)

OTHER COMMENTS:
- line 35:         replace "computes" by "identifies
- line 73:         can't one make the argument that each example in a sequence is a pool of size 1?
- line 81:         can --> may/will
- line 81:         please point to the exact section/paragraph/experiment that illustrates the detrimental effect
- line 158:       please provide details on unstable behavior
- line 174:       please quantify HOW "MUCH MORE"
- line 223:       this is a bit "out of the blue"; please provide details or remove (as it is highly speculative, and does not belong in a advantages/limitations section)
- lines 7/12/200-202 make this approach similar in spirit with the "aggressive co-testing" one in the paper below (including the value of detecting the outliers); while your approach is completely original, the similarity may be worth noting.

Muslea, I., Minton, S., & Knoblock, C. A. (2006). Active learning with multiple views. Journal of Artificial Intelligence Research, 27, 203-233.

**Time Spent Reviewing:**

3

---

> ### Author Response · Authors · 2021-08-10
> **Reply to reviewer CVDx**
>
> Thank you for your insightful and encouraging review.
>
> > What proportion of the mislabeled examples can you detect?
>
> CINCER detects around 75% of the mislabeled examples compared to 50% of the other methods. Figure 4 in the paper reports the total number of cleaned examples.
>
> > What proportion of the detected contrastive pairs do NOT contain at least one example for which you did NOT corrupt the label?
>
> Good question.  This corresponds to the case of "useless" queries.  Our experience is that around 5% of the queries are useless in this sense, i.e., neither the suspicious example nor the counter-example are corrupted.
>
> > Adding to Figures 2 & 4 the upper-bound performance of the model.
>
> These are all very valid points.  We will update the plots in the supplementary material to integrate this information.

---

> ### Comment · Reviewer_CVDx · 2021-08-30
> **Maintaining original rating "7: Good paper, accept"**
>
> Thank you for proving clear answers to my comments and questions. The paper still is a solid ACCEPT.

---

### Official Review · Reviewer_ikcA · 2021-07-16

**Rating:** 8
**Confidence:** 4

**Summary:**

The paper tackles the problem of label cleaning, where labels for incoming data points might be flawed. The authors propose an interactive learning approach, where the system builds counter examples for data points, which consist of pairs of data points which the model finds incompatible. The human expert is then asked to relabel either or both of these presented data points.

The empirical evaluation applies the method to tabular- and image data, comparing the proposed approach with skeptical learning and a variation of the proposed approach without counter examples. The results show that the approach outperforms its competitors in terms of F1 score. The authors also validate that using the fisher information matrix supports finding good counter examples.

**Limitations And Societal Impact:**

The authors addressed potential negative societal impact in the conclusion. The authors could additionally address limitations of the system in terms of potential (special) situations where the approach might break.

**Main Review:**

Positive points:
* The paper is well-written and has clear design decisions
* Although the paper reuses ideas, the resulting application is sufficiently novel and original

Negative points:
* Empirical evaluation could ablate margin parameter and use more challenging data sets

The paper proposes an interesting and well-motivated solution to label cleaning. The approach conceptually extends the setting of cleaning novel data points by also enabling to clean the available dataset, which is sensible in online settings with little to no initial knowledge.

The solution to finding expressive counter examples for expert labeling is also well-motivated, building on the concept of influence functions and fisher information. The paper also contributes with transferring the use of influence functions to the online setting.

With respect to the elaborated related works, I only have one comment/question. The provided list of papers and explained differences are sensible, however: the authors state that LWA assums a pool-based setting, where data is not perceived fully online. Could you please elaborate why label cleaning requires the full online setting? It would be good to thoroughly motivate it, as it is stated as main reason to build on / compare against skeptical learning only. I understand that the training set would not be built fully sequentially in the pool-based setting, but could one repeatedly use LWA on smaller batches?

I find the evaluation to be convincing, as diverse data sets have been used and the method has been partially ablated and compared against skeptical learning. It would, however, also be interesting to test approach on challenging image datasets (other than MINST-based datasets), but it is sensible to start with those. Also, did you also ablate the margin threshold parameter? What would happen if one decrease/increased the parameter? How easy is it to tune it for new data?

Lastly, I was wondering in what (adversarial) cases the approach could break in the online setting. Could the approach converge to not asking the expert sufficiently often or to never find a large portion of noisy labels? If yes, could one categorize such cases? What if the amount of noise is very large?

### Update after author response:
I thank the authors for the thorough answers to my questions! After reading the other reviews, my assessment of the paper remains utterly positive.


**Time Spent Reviewing:**

4 hours

---

> ### Author Response · Authors · 2021-08-10
> **Reply to reviewer ikcA**
>
> Thank you for your thoughtful and encouraging review.
>
> > Could you please elaborate why label cleaning requires the full online setting?
>
> We focused on sequential learning because this is how our key use case (smart personal assistants) work.  Our counter-example based strategy can definitely be applied in offline and pool-based active learning tasks too.
>
> > Test approach on challenging image datasets
>
> One issue with this scenario is that prediction in complex datasets requires training with many examples, which makes these settings not representative of the kind of interactive tasks that we are interested in.  More complex datasets will be considered for a longer version of the paper and/or for an offline/crowd-sourced version of CINCER.
>
> > Did you also ablate the margin threshold parameter? What would happen if one decrease/increased the parameter? How easy is it to tune it for new data?
>
> The threshold was chosen during initial experiments to ensure that there was a clear difference between skeptical and non-skeptical online learning (regardless of counter-examples) and then kept fixed across competitors to make comparison easier.
>
> The threshold balances between interacting with the user too often (at the cost of cognitive effort)  and not enough (which in turn would lower the benefits of our counter-example strategy).
>
> Tuning parameters in online settings is not entirely straightforward and, to the best of our knowledge, there is no framework to select them automatically.  One option is to let the user choose whether it is receiving too many questions and to monitor the model’s performance on an (incrementally grown) validation set to balance the threshold appropriately;  this would require a thorough separate validation.  This is a very interesting and useful research direction, but orthogonal to our contribution.
>
> > I was wondering in what (adversarial) cases the approach could break in the online setting. Could the approach converge to not asking the expert sufficiently often or to never find a large portion of noisy labels? If yes, could one categorize such cases? What if the amount of noise is very large?
>
> The approach can break under at least two conditions -- excessive amount of noise and substantial amounts of systematic noise, as suggested by reviewer hs1w -- for different reasons.
>
> On the one hand, too much label noise makes the model uncertain everywhere, making the skeptical noise fire all the time.  On the other, systematic noise can also hide future mislabelled examples falling close to regions that host systematic noise from the skeptical check, because in those regions is certain that the noisy label is actually correct.  (This was briefly mentioned in the Introduction.)
>
> Notice however that the main issue lies in the skeptical check, which we took from the literature, not in our counter-example selection strategy. As such, improvements to the skeptical component are likely to help in this regard.
>
> An alternative is to instead leverage a multi-round counter-example strategy, such that once a mislabeled counter-example is discovered, its (systematically mislabelled) counter-counter-examples can be recovered and fixed by iterating our counter-example selection strategy, this time w.r.t. the counter-example.  We will make sure to clarify these points in the text.

---

### Official Review · Reviewer_j1iQ · 2021-07-16

**Rating:** 8
**Confidence:** 4

**Summary:**

The paper introduces an approach called CINCER which tackles label noise in a sequential learning task by querying a human annotator whenever it finds a suspicious example. In support of its suspicion, the model also presents the user a maximally-incompatible counter-example from the already accumulated training set. The primary contribution of the paper lies in choosing an informative counter-example using Fischer information matrix as an approximation of influence functions. CINCER enables the user to correct both incoming and old examples.

**Limitations And Societal Impact:**

Yes, the authors have addressed the limitation of their works remarkably well.

**Main Review:**

The paper is well-written with adequate representation given to prior work. I would, however, like the authors to include the highly-relevant and recent work on Confident learning in their literature work (Northcutt, Curtis, Lu Jiang, and Isaac Chuang. "Confident learning: Estimating uncertainty in dataset labels." Journal of Artificial Intelligence Research 70 (2021): 1373-1411.).

The idea of presenting the human annotator with a maximally-incompatible example as a means to explain the model's suspicion is an exciting idea. The authors admit that the counter-examples produced by CINCER may not be perceptually similar, which may impact the their explainability. However, I agree that future work may focus on some reasonable tweaks to incorporate that aspect. Another recommendation is having the ability to incorporate batching since it seems inefficient if a model is trained every time a new training example is obtained.

I feel that the claim that CINCER can also clean the already accumulated dataset is slightly stretched, primarily because the discovery of old mislabelled data is somewhat driven by the new incoming data. Therefore, there's a significant chance that some of the old mislabelled data may never come to light. In this regard, the experiments are also a bit unclear to me. For instance the paper mentions that "20% of the training examples are corrupted at random" and "before each run, the models are trained on an bootstrap training set of 100/500 examples". It is unclear as to whether (1) the bootstrap training set has mislabelled data points, and (2) what is the proportion of mislabelled data in the initial bootstrap sample. Explicit control on the noise in the bootstrap might allow us to test the model's ability to reveal mistakes in the "old" dataset. Since the paper claims that this is a key differentiating factor from prior work on skeptical learning, I would be interested in seeing such experiments.

In Section 4.1 Q1, I believe it would be interesting to have NN as another baseline because from Section 4.2 it doesn't seem to do significantly poorly.

**Time Spent Reviewing:**

2

---

> ### Author Response · Authors · 2021-08-10
> **Reply to reviewer j1iQ**
>
> Thank you for your thoughtful and encouraging review.
>
> > Northcutt, Curtis, Lu Jiang, and Isaac Chuang. "Confident learning: Estimating uncertainty in dataset labels." JAIR (2021).
>
> Thank you for this very relevant reference, we will make sure to discuss it in the text.
>
> > having the ability to incorporate batching since it seems inefficient if a model is trained every time a new training example is obtained.
>
> This batch-based setup does not immediately apply to our sequential learning setting and we did not consider online settings with delayed updates (in which the model is retrained after every n examples).  Regardless, CINCER can definitely be adapted to this setup and even to batch-based/deep active learning.
>
> This actually opens an interesting opportunity: depending on the size of the batch and on the application, it might make sense to only require counter-examples for the ``most jointly suspicious’’ examples in a batch.  In particular, it would be reasonable to choose subsets of suspicious examples that can *all* be clarified by cleaning up a *single* mislabeled counter-example (i.e., a set of suspicious examples with a common root cause), thus reducing the total annotation effort.  Thank you for suggesting this direction, this is a very interesting avenue for future work.
>
>
> > there's a significant chance that some of the old mislabelled data may never come to light.
>
> This is correct: the portion of the training set cleaned at runtime depends on the distribution of the incoming examples.  One could however argue that noisy training examples that do not affect the margin on future examples do not require urgent cleaning, and that even these examples will likely be cleaned once future instances reveal their presence to the skeptical check.  This “lazy” approach is actually useful in interactive pipelines as it helps to limit the user’s effort.  We will make sure to rephrase the text so as to avoid any confusion.
>
> > It is unclear as to whether (1) the bootstrap training set has mislabelled data points, and (2) what is the proportion of mislabelled data in the initial bootstrap sample.
>
> This is the setting that we used in the experiments: the bootstrap set has exactly 20% noise. We will clarify this in the text.
>
> > In Section 4.1 Q1, I believe it would be interesting to have NN as another baseline because from Section 4.2 it doesn't seem to do significantly poorly.
>
> The plots for Q1 and Q3 report different curves/competitors taken from the same underlying experiment, purely for readability, and can be compared directly.

---

> ### Comment · Reviewer_j1iQ · 2021-08-31
> **Stand by my original rating**
>
> I thank the authors for clearly answering my questions. Excellent work!

---

### Official Review · Reviewer_hs1w · 2021-07-16

**Rating:** 6
**Confidence:** 4

**Summary:**

The paper introduces a new data cleaning technique (CINCER), which selects potentially mislabeled samples to present to an expert for relabelling. CINCER works by identifying so-called `suspicious' samples and selecting a counter-example for each of them. The suspicious sample and the counter-example are both provided to the human expert for re-annotation. Samples are selected by maximizing the margin for a probabilistic classifier, while counter-example selection is performed via an approximation of the influence function which uses the Fisher information matrix. The use of the approximation is theoretically justified (mostly by pointing to relevant work), and experimentally shown to be a better choice than the alternatives from either the perspective of computational cost or performance (data cleaning / F1 score of resulting model).

**Limitations And Societal Impact:**

The limitations of the work are addressed. In terms of negative societal impact, this work is on par with other data cleaning techniques requiring the intervention of human users -- for instance, it is vulnerable to malicious annotators. However, this is due to the nature of the task, not a drawback of the paper itself.

**Main Review:**

The paper introduces a new and pertinent way of selecting samples for label cleaning. The method is clearly presented, with every step being explained, most importantly the need for approximating the influence function. The choice of approximations using the FIM does rely on past work, albeit its use in this context is ingenious. The authors are also as upfront about the (unavoidable) limitations of their approach as they are of the advantages. The authors have also nicely structured the experiments in support of each of their hypotheses, making section 4 clear and compelling. Overall, the method is sufficiently sophisticated and of demonstrated usefulness to be of interest to the community.

A weak point of the paper is related to the claims around interpretability: The authors indicate that the counter-examples serving as 'explanations of the model's suspicions'. This isn't strictly speaking the case, according to many percepts of explainability/interpretability. While the counter-examples could provide some clues as to why the model might have made a mistake (or, conversely, point to a labeling error), it is far from providing a proper explanation (an example of an explanation, in the case of figure 1, would be 'the sample was selected because itt was labeled as a 9 yet there's a gap in the top part').  This doesn't diminish what the method actually achieves, however, statements that CINCER explains `the reasons behind the model’s skepticism' should be toned down, to avoid over-claiming wins on the interpretability side, especially since no actual user study was conducted.

Another limitation of the paper is that all the data corruption was performed in the same way (20% of training samples were corrupted at random in all datasets). This, of course, means that mislabeled samples are relatively frequent (thousands, in some cases, out of which the model cleans around 20-30). What happens if the mislabeled samples aren't quite that easy to find? (i.e. the percentage of mislabeled samples is lower) What happens if the mislabeling is systematic?
It would be interesting to see how many of the cleaned samples are 'counter-examples' and how many are the suspicious samples themselves as an indication of how effective CINCER is in dealing with past data.

Needless to say, the experimental section would be a lot stronger if a real  crowdsourced dataset was selected, with real human expert reviewers to correct the labels. If two labels are provided per iteration and there are a hundred of iterations, such a study is perfectly doable.


Clarity/presentation issues:

- Figure 1 mentions counter-examples being 'contrastive and influential' when neither of these terms have been clearly defined yet.

- the related work in section 5 is somewhat repetitive of the introduction

- typo: "details depends"

**Time Spent Reviewing:**

5

---

> ### Author Response · Authors · 2021-08-10
> **Reply to reviewer hs1w**
>
> Thank you for your insightful and encouraging review.
>
> > statements that CINCER explains `the reasons behind the model’s skepticism' should be toned down, to avoid over-claiming wins on the interpretability side
>
> We believe explainability to be an integral part of our method, for two reasons.  Explanations paint a necessarily simplified picture of the mechanisms underlying a specific decision.  Example-based explanations, in particular, hide attribute relevance information.  This does not disqualify them as explanations.  This view is supported by the many works on example-based explanations in XAI, including [12,16,20,22,26,36,45] and the many others on prototypes and exemplars.  Second, our “pertinence” desideratum (D3) has the specific aim of keeping our interaction protocol interpretable;  of course, more could be done, as we discussed right after D3 in the text.
>
> We completely agree that attribute-level explanations provide important and complementary information, and indeed we mentioned the possibility of integrating them into our approach in the Related Work, in “Other works”.  We will give more visibility to this point.
>
> > What happens if the mislabeled samples aren't quite that easy to find?
>
> We did carry out some preliminary experiments with different noise levels on MNIST.  Initial results showed that CINCER performs similarly as in the results that we reported, at least in terms of percentage of cleaned examples.  We settled for 20% noise because for lower percentages the noise had relatively little effect on the F1 for our models on MNIST, rendering cleaning almost irrelevant.
>
> > What happens if the mislabeling is systematic?
>
> Good question.  Pre-existent label noise can fool the skeptical check, because “in the neighborhood” of a noisy example the machine is quite sure that the noisy label is correct (i.e., the margin is large);  hence, it will rarely be skeptical about future noisy examples falling in that region.  Systematic noise exacerbates this situation.
>
> This issue is unrelated to the counter-example selection strategy, though, so one option for dealing with systematic noise is to make the skeptical check more robust.
>
> An alternative -- more in line with our proposal -- is to leverage a *multi-round* strategy, as mentioned in the conclusion.  The idea is that, once a mislabeled counter-example is discovered, a (systematically mislabelled) "counter-counter-example" can be recovered and fixed by applying our IF strategy to the counter-example.
>
> This could be annotation heavy.  However, if the shape of the noise is known, more efficient counter-example cleaning strategies (that, e.g., select and clean *groups* of systematically mislabelled examples) could be devised.
>
> We will include a short discussion of systematic noise in the text.
>
> > How many of the cleaned samples are counter-examples/suspicious samples?
>
> We added this information to the supplementary material.  The new curves show that CINCER tends to clean more suspicious examples than counter-examples, in a ratio of roughly 2 : 1.  Compared to DropCE, CINCER corrects roughly the same number of counter-examples but many more suspicious examples.  This further shows that cleaning the training set improves the ability of the model of being ``suspicious for the right reasons’’.
>
> > The experimental section would be a lot stronger if a real crowdsourced dataset was selected, with real human expert
>
> We completely agree.  An experiment with human subjects is currently in the planning stage for a longer version of the paper.
>
> > Vulnerable to malicious annotators
>
> Thank you, we had not considered this case.  We will update the potential negative impact paragraph accordingly.

---

### Decision · Program_Chairs · 2021-09-27

**Decision:**

Accept (Spotlight)

**Comment:**

Four knowledgeable referees have thoroughly reviewed this paper and all recommended that it is accepted. I agree with their recommendation.